# Is Sitting Always Inactive and Standing Always Active? A Simultaneous Free-Living activPal and ActiGraph Analysis

**DOI:** 10.3390/ijerph17238864

**Published:** 2020-11-28

**Authors:** Roman P. Kuster, Wilhelmus J. A. Grooten, Victoria Blom, Daniel Baumgartner, Maria Hagströmer, Örjan Ekblom

**Affiliations:** 1Institute of Mechanical Systems, School of Engineering, ZHAW Zurich University of Applied Sciences, 8400 Winterthur, Switzerland; daniel.baumgartner@zhaw.ch; 2Division of Physiotherapy, Department of Neurobiology, Care Sciences and Society, Karolinska Institutet, 141 83 Huddinge, Sweden; wim.grooten@ki.se (W.J.A.G.); maria.hagstromer@ki.se (M.H.); 3Medical Unit Occupational Therapy and Physiotherapy, Allied Health Professionals, Karolinska University Hospital, 171 77 Stockholm, Sweden; 4Department of Physical Activity and Health, The Swedish School of Sport and Health Sciences, 114 86 Stockholm, Sweden; victoria.blom@gih.se (V.B.); orjan.ekblom@gih.se (Ö.E.); 5Division of Insurance Medicine, Department of Clinical Neuroscience, Karolinska Institutet, 171 77 Stockholm, Sweden; 6Academic Primary Health Care Center, Region Stockholm, 104 31 Stockholm, Sweden

**Keywords:** active sitting, bland-altman, inactive standing, method comparison, posture and physical activity index (POPAI), sedentary behavior

## Abstract

Sedentary Behavior (SB), defined as sitting with minimal physical activity, is an emergent public health topic. However, the measurement of SB considers either posture (e.g., activPal) or physical activity (e.g., ActiGraph), and thus neglects either active sitting or inactive standing. The aim of this study was to determine the true amount of active sitting and inactive standing in daily life, and to analyze by how much these behaviors falsify the single sensors’ sedentary estimates. Sedentary time of 100 office workers estimated with activPal and ActiGraph was therefore compared with Bland-Altman statistics to a combined sensor analysis, the posture and physical activity index (POPAI). POPAI classified each activPal sitting and standing event into inactive or active using the ActiGraph counts. Participants spent 45.0% [32.2–59.1%] of the waking hours inactive sitting (equal to SB), 13.7% [7.8–21.6%] active sitting, and 12.0% [5.7–24.1%] inactive standing (mean [5th–95th percentile]). The activPal overestimated sedentary time by 30.3% [12.3–48.4%] and the ActiGraph by 22.5% [3.2–41.8%] (bias [95% limit-of-agreement]). The results showed that sitting is not always inactive, and standing is not always active. Caution should therefore be paid when interpreting the activPal (ignoring active sitting) and ActiGraph (ignoring inactive standing) measured time as SB.

## 1. Introduction

Sedentary Behavior (SB) has become an emerging research field. A large number of studies associated sedentary time and more recently prolonged sedentary time (with a minimum bout duration) and the sedentary accumulation pattern with chronic lifestyle diseases and premature deaths [1,2,3,4,5,6]. To measure SB reliably and accurately, it is well established that sensor-based methods should be employed, and accelerometers are the method of choice [7,8]. Due to different placement and data processing, the accelerometers can be separated in two types: sensors to measure posture (also known as inclinometers) and sensors to measure physical activity (also known as movement sensors). Posture sensors low-pass filter the acceleration signal to determine the sensor’s orientation versus gravity [9]. Attached to the thigh, they provide an accurate estimate of the time spent sitting and standing [10]. The most commonly used posture sensor is the activPal (PAL Technologies, Glasgow, UK), for which the measured sitting time is taken as a direct estimate of SB [11]. The second type, physical activity sensors, traditionally convert the raw acceleration signal (in m/s^2^) into a counts-per-minute (cpm) measure to describe the intensity of the sensor’s movement within a minute [12]. Attached to the waist, they provide a reasonable valid estimate of the time spent in different physical activity levels [13]. While the cut-points to detect SB differ between sensors, placements and study populations, the most commonly used activity sensor is the ActiGraph GT3X (ActiGraph LCC, Pensacola, FL, USA), for which the time spent below 100 cpm on the vertical axis is usually taken as a direct estimate of SB [14].

However, the latest and most widely used definition of SB requires a certain body posture (sitting or reclining) plus a certain physical activity level (≤1.5 metabolic equivalents) [15]. Therefore, the common practice of measuring SB with just one sensor, either an activPal (posture) or an ActiGraph (physical activity), represents a serious limitation. Aware of this fact, numerous studies compared the sedentary estimates of the two sensors, and reported quite similar results [13,16,17]. However, this does not mean that the two sensors measure the same behavior nor does it mean the sensors have a good validity to measure SB. It only means that the behavior measured by the two sensors are similar common. The degree to which the sedentary bouts of the two sensors truly overlap is still unknown, and requires a simultaneous, time-matched analysis with both sensors. In fact, the activPal does not measure SB but sitting, and the ActiGraph does not measure SB but minimal-intensity physical activity (minPA, defined by a metabolic equivalent ≤1.5) [7]. Consequently, only the time that both sensors classify simultaneously as SB truly complies with the definition. The activPal’s measured sitting time not classified as minPA by the ActiGraph should be classified as active sitting, and the ActiGraph’s measured minPA time classified as standing by the activPal should be classified as inactive standing. Up to now, it remains unknown how frequent active sitting and inactive standing occurs in everyday life, and thus to which degree this affects the estimated sedentary time (with and without a minimum bout duration) and the sedentary accumulation pattern of the two sensors.

The aim of this study was to determine the true amount of active sitting and inactive standing in daily life, and to analyze by how much the two behaviors falsify the sedentary estimates and the sedentary accumulation pattern of the activPal and ActiGraph compared to a combined sensor analysis, the Posture and Physical Activity Index (POPAI). We expected that both sensors significantly overestimate sedentary time, and that the activPal bias depends on the amount of active sitting, and the ActiGraph bias depends on the amount of inactive standing.

## 2. Materials and Methods

To determine the true amount of active sitting and inactive standing in daily life, the study described the daily wake-time use of office workers with the combined posture and physical activity classification of POPAI. Furthermore, to determine by how much active sitting and inactive standing falsify the sedentary estimates and the sedentary accumulation pattern of the activPal and the ActiGraph, Bland-Altman statistics compared the single sensor estimates to the combined sensor estimate of POPAI. 

### 2.1. Participants

This study aimed for a sample size of 100, which means that the 95% confidence interval of the Bland-Altman bias equals approximately one third of the standard deviation of the difference between the methods [18]. Considering 5% sensor malfunction and non-wear, an initial sample of *n* = 105 was used. All participants were recorded within the Brain-Health-Study investigating the association between physical activity pattern and cognition, mental health, and sleep in office workers [19]. The study protocol was approved by the regional ethics board Stockholm (ID 2016/796-31), and all participants signed an informed consent prior to study inclusion.

### 2.2. Data Recording

Both sensors were initially mounted by the research staff, and participants were instructed to wear them for at least 7 days. The activPal was worn continuously on the right thigh (attached with a waterproof tape), and the ActiGraph GT3X on a belt around the waist during waking hours and around the wrist during sleep (Figure 1). For the present study, only waking hours with waist-worn ActiGraph were analyzed. Participants kept a diary to note ActiGraph waist-worn time.

### 2.3. Valid Time Detection

To limit the behavior classification to waking hours with valid data from both sensors, a similar processing as described in Kuster et al. 2020 was used (see Appendix A for a step-by-step instruction on how the data were processed). In short, the processing consisted of four steps: (1) identify valid activPal days; (2) synchronize sensor data; (3) exclude ActiGraph non-wear time; and (4) remove short episodes and limit data to days with ≥10 valid hours [17].

Valid activPal days contained at least 500 steps, 12 h (without bedtime), and <95% of the time spent in one activPal code [20]. Bedtime was removed using an automated activPal algorithm [20]. As the algorithm slightly underestimates bedtime, days with bedtime start after 1:00 am or end before 4:30 am were visually inspected and adjusted if required using the diary information. To synchronize the sensor data, the offset of the two sensor clocks was determined on the raw data and applied to the ActiGraph time. A detailed discussion of the asynchronous sensor clocks can be found elsewhere [17]. In short, the raw data comparison of the two sensors showed an obvious temporal mismatch. Episodes with large acceleration values typically started a couple of seconds delayed on the ActiGraph compared to the activPal, and the delay increased over time (concrete example given in Appendix A).

ActiGraph non-wear time was then excluded by inspecting all activPal events overlapping an ActiGraph episode of at least 30 s with constant raw signal (i.e., no sensor movement at all), and removing those for which one of the following criteria was true: (1) the activPal reported a posture change; (2) the activPal classified part of the episode as stepping; (3) the ActiGraph episode lasted ≥ 90 min. Criteria 1 and 2 took into account that it is practically impossible to measure an active behavior with one sensor (activPal) without any raw signal change on the other (ActiGraph), and criterion 3 took into account that keeping a static posture without any raw signal change for 90 min is very unlikely. Last, to prevent excessive fragmentation of the data, short episodes in between longer excluded episodes were removed, and only days with ≥10 valid hours after all these exclusions were kept in the analysis.

### 2.4. Behavior Classification

#### 2.4.1. activPal

For the activPal classification, the event file (csv generated with activPal3 v7.2.38) with the proprietary behavior classifications “sedentary”, “standing”, and “stepping” was used. Note that “sedentary” was renamed to “sitting” to clarify that only the posture component of SB is considered.

#### 2.4.2. ActiGraph

For the ActiGraph classification, the 1-s count file with low-frequency-extension filtering (csv generated with ActiLife v6.13.4) was aggregated into minute-by-minute data, and each minute was classified as SB, light-intensity physical activity (LIPA), or moderate- to vigorous-intensity physical activity (MVPA). Subsequent minutes with the same activity classification were summarized into bouts. Vertical axis cut-points of 100 cpm and 1952 cpm were used to separate “SB”, “LIPA”, and “MVPA”. These two cut-points are, according to Migueles et al. 2017, those most often used to separate SB, LIPA, and MVPA with the ActiGraph GT3X [14]. Note that “SB” was renamed to “minPA” to clarify that only the physical activity component of SB is considered.

#### 2.4.3. POPAI

The combined posture and physical activity analysis started with the activPal event file, and classified each minute of an event into the corresponding activity level using the time-matched ActiGraph 1-s counts (Figure 2). The activPal category sitting was thus split into “inactive sitting” (compliant with the definition of SB) and “active sitting”, and the activPal category standing was split into “inactive standing” and “active standing”. In case the activity classification changed during an activPal event, the event was split accordingly. An exemplary classification of a 20-min recording is given in Figure 2, whereas the first 5 min show an artificial minute-based behavior to demonstrate the combination of activPal and ActiGraph data into POPAI, and the remaining 15 min show a natural non-minute-based behavior. Informed by a previous study comparing the ActiGraph activity classification for minPA and LIPA in sitting and standing to an indirect calorimeter, posture specific cut-points were used (75 cpm to separate inactive and active sitting, 150 cpm to separate inactive and active standing) [21]. Posture events <1 min were classified with the corresponding fraction of the cut-point (e.g., 56 cpm instead of 150 cpm for a 22.4-s standing event, Figure 2), and events <1 s were ignored (no matching ActiGraph count). When looking from an activity perspective on the POPAI behavior classification, inactive sitting and inactive standing are both minPA, and active sitting and active standing are both LIPA. For the present analysis, activPal “stepping” was kept as an own category to keep the total time for each method constant. Stepping was considered active without any further separation into LIPA and MVPA.

### 2.5. Data Analysis and Statistics

To account for the inconsistent daily waking hours, the time spent in each behavior was expressed relative to waking hours and averaged for each participant. Descriptive time use data for each method is, after failing to reject the normal distribution assumption with Lilliefors test, presented as pie chart with mean ± standard deviation and 5th to 95th percentile.

Total sedentary time as well as the time spent in prolonged sedentary bouts of ≥10 min and ≥30 min a day was compared with Bland-Altman Statistics between the methods, with POPAI as the reference. The comparison used the bias with 95% confidence interval as measure of accuracy and the 95% limit of agreement as measure of precision [22]. In case the difference between the methods depended on the average of both methods (tested as outlined in [22] with *p* ≤ 0.05), the linear regression approach was used and the bias and 95% limit of agreement is presented at the mean of both methods. The bias was considered significant if its 95% confidence interval excluded zero. As we expected that the activPal bias depends on the amount of active sitting and the ActiGraph bias depends on the amount of inactive standing, we additionally plotted the difference in total sedentary time between the methods against POPAI measured active sitting and inactive standing, and calculated the linear correlation. The correlation is presented with the squared Pearson correlation coefficient including the 95% confidence interval.

The sedentary accumulation pattern was calculated for each method and likewise compared with Bland-Altman statistics. Informed by a recent review on this topic [23], the following pattern variables were used: (1) number of sedentary bouts a day: for total, ≥10-min and ≥30-min bouts; (2) median bout length; (3) percentage of time spent in bouts of at least the median bout length; (4) half-life bout duration: the bout length at which 50% of sedentary time is accumulated [24]; and (5) Gini coefficient: a coefficient ranging from 0 to 1, with 0 indicating that all bout lengths contribute equally to the total sedentary time, and 1 indicating that total sedentary time is dominated by the longest sedentary bout [25,26]. Note that the number of sedentary bouts a day was calculated on a day-by-day level and averaged for each participant, and the remaining pattern variables were calculated over the entire recording of each participant.

## 3. Results

From the 105 participants enrolled in the analysis, five provided no valid day and were excluded from the analysis. The remaining 100 participants were on average 40.8 ± 9.2 years old (range: 21–64) and weighed 71.6 ± 12.6 kg (range: 49.2–104.0). In total, 725 days with an average recording time of 15.0 ± 0.8 h were analyzed. The initial offset that was corrected between sensor clocks averaged 6.3 (9.2) [−2.7–25.8] seconds and increased by 1.0 (1.3) [0.0–2.8] seconds per day (median with inter-quartile range and [5th–95th percentile]). The positive offset indicates an ActiGraph delay.

### 3.1. Descriptive Time Use

Descriptive data of the daily wake time use is shown for each method in Figure 3. POPAI showed that 23.9 ± 7.0% [14.6–35.6%] of the total sitting time was spent in LIPA (active sitting), and 41.3 ± 12.6% [22.1–63.5%] of the total standing time was spent in minPA (inactive standing, mean ± standard deviation [5th–95th percentile]). Furthermore, 32.5 ± 7.8% [21.0–46.9%] of the total active time (LIPA and MVPA) was accumulated in sitting, and 21.3 ± 8.7% [10.0–38.3%] of the total minPA time was accumulated in standing.

### 3.2. Sedentary Time

Both, the activPal and the ActiGraph significantly overestimated total sedentary time by 13.6% and 10.1% of the waking hours, respectively (Table 1, Figure 4). Relative to POPAI, the activPal overestimated total sedentary time by 30.3% [28.5%–32.1%] and the ActiGraph by 22.5% [20.6%–24.5%] (bias with 95% confidence interval). The bias did not depend on total sedentary time, but on total active sitting and inactive standing time (Figure 5). Active sitting explained 100% [100%–100%] of the activPal bias, and inactive standing explained 92.3% [88.7%–94.8%] of the ActiGraph bias (r^2^ with [95% confidence interval]).

Both sensors also significantly overestimated prolonged sedentary time accumulated in bouts of ≥10 min and ≥30 min (Table 1, Figure 4). Interestingly, the bias for the activPal was, relative to waking hours, larger for prolonged than for total sedentary time (≥20.0% versus 13.6%), and the bias for the ActiGraph was, relative to waking hours, lower for prolonged than for total sedentary time (≤6.7% versus 10.1%). However, relative to POPAI, both sensors had a larger overestimation for prolonged than for total sedentary time (83% and 233% for the activPal and 25% and 36% for the ActiGraph).

### 3.3. Sedentary Accumulation Pattern

Both sensors significantly deviated from POPAI for all variables except the ActiGraph for the Gini-Coefficient (Table 2). Regardless of a minimum bout duration, the number of sedentary bouts a day were overestimated by both sensors, except for the activPal, which underestimated the total number of sedentary bouts a day. The median bout length was overestimated by the activPal and underestimated by the ActiGraph, but the percentage of time spent in bouts of at least the median bout length was overestimated by both. Furthermore, the bout length at which 50% of sedentary time is accumulated (half-life bout duration) was overestimated by both sensors. Last, the Gini-coefficient, a coefficient ranging from 0 to 1, with 0 indicating that all bout lengths contribute equally to total sedentary time and 1 indicating that total sedentary time is dominated by the longest sedentary bout, was overestimated by the activPal but not by the ActiGraph.

## 4. Discussion

The present study analyzed the daily time spent inactive sitting (equal to SB), active sitting and inactive standing, and the degree by which measuring only one component of SB, either posture or physical activity, falsifies the SB estimates and accumulation pattern. The results showed that it seriously matters how SB is measured. Both, the activPal and the ActiGraph substantially overestimated sedentary time compared to the combined posture and physical activity classification (POPAI), and neither of the single sensors can be recommended to measure sedentary time compliant with its definition. The reason for this is that sitting is not always inactive and standing is not always active. In fact, the investigated sample spent 24% of the sitting time active (LIPA), and 41% of the standing time inactive (minPA). This also means that 33% of the active time (LIPA and MVPA) was accumulated in sitting, and 21% of the inactive time (minPA) was accumulated in standing. Some participants even spent most of their standing time inactive (95th percentile equals 64%), while others accumulated almost half of their active time (LIPA and MVPA) in sitting (95th percentile equals 47%).

### 4.1. Sedentary Time

Total sedentary time a day estimated with the activPal (58.6% of the waking hours or 8.8 h a day) and the ActiGraph (55.1% of the waking hours or 8.3 h a day) were in a similar range, but POPAI showed a substantial temporal miss-match between the two sensors. Participants actually spent 13.7% of the waking hours or 2.0 h a day active sitting (activPal sitting and ActiGraph LIPA), as well as 12.0% of the waking hours or 1.8 h a day inactive standing (activPal standing and ActiGraph minPA). Accordingly, the sedentary estimate of the combined analysis (45% of waking hours or 6.7 h a day) was substantially lower, and the activPal and ActiGraph overestimated sedentary time by almost one third and one quarter, respectively. The overestimation was, relative to POPAI, even larger when applying a minimum bout length (up to 233%). The substantial overestimation indicates a low accuracy of the single senor methods to measure sedentary time, with and without applying a minimum bout duration. This would not be a serious issue in case the bias is constant among the individuals (i.e., high precision), but the 95% limit of agreements (Table 1 and Figure 4) showed a substantial spread of the bias among the individuals, indicating a limited precision. For total sedentary time, the activPal bias depended fully on the amount of active sitting (r^2^ = 1.0), and the ActiGraph bias depended strongly on the amount of inactive standing (r^2^ = 0.94), but neither bias depended on the amount of inactive sitting itself (Figure 5).

### 4.2. Sedentary Accumulation Pattern

The comparison of the sedentary accumulation pattern shows different pictures for the two sensors on how SB was accumulated. The activPal underestimated the total number of sedentary bouts despite overestimating total time spent sedentary. This likely results from the fact that one long activPal sitting bout might become several shorter POPAI inactive sitting bouts in case the sitting bout contains some LIPA minutes (Figure 2, last activPal bout). This means that the activPal underestimates the total bout number but overestimates the total time. Since some of the resulting shorter inactive sitting bouts do not fulfil the minimum prolonged bout duration, the number and time of prolonged bouts was overestimated by the activPal. Consequently, the activPal overestimated the median bout length, the percentage of time spent in bouts of at least the median bout length, the half-life bout duration, and the Gini-Coefficient, all indicating that longer bouts contribute to a larger degree to the total sedentary time.

On the other hand, the ActiGraph overestimated the time spent sedentary as well as the number of sedentary bouts. This observation can be explained by the fact that the ActiGraph neglects the posture component and counts inactive sitting as well as inactive standing bouts as sedentary (Figure 2, minute 3 and 11). The fact that the number and time of prolonged bouts was overestimated by the ActiGraph indicates that some prolonged minPA bouts contained a posture change, splitting up inactive sitting (POPAI) but not minPA (ActiGraph). Due to the inclusion of inactive standing, the ActiGraph underestimated the median bout length. However, the ActiGraph only slightly overestimated the percentage of time spent in bouts of at least the median bout length, the half-life bout duration, and the Gini-Coefficient (which was actually the only pattern measure non-significantly different from POPAI).

### 4.3. Critical POPAI Appraisal

POPAI is not the first attempt to combine the data processing of two sensors to improve the measurement of SB, nor is the present study the first to compare the isolated posture and physical activity classification to a combined posture and physical activity classification. Ellingson and colleagues (2016) combined a machine-learning algorithm (sojourn) for a waist-worn ActiGraph with a thigh-worn activPal to refine the algorithms posture transitions and classify inactive bouts into sitting and standing [27]. Myers and colleagues (2017) combined a minute-based posture classification of a thigh-worn activPal with the SenseWear Armband (BodyMedia, Inc., Pittsburgh, PA, USA) to identify the wake time (SenseWear) spent sitting (activPal) and in minPA (SenseWear) [28]. Fanchamps and colleagues (2017) used the 3-sensor Vitaport activity monitor (TEMEC, Kerkrade, The Nederland) with two thigh-worn and one trunk-worn accelerometers to identify the sitting time spent in minPA [29]. However, compared to these studies, we see three major advantages of the sensor combination presented here: (1) POPAI combines the two most common sensors to measure SB, a thigh-worn activPal and a waist-worn ActiGraph; (2) the combination uses the well-established proprietary data processing of both sensors, including the event based posture classification of the activPal; and (3) POPAI allows to have posture-specific ActiGraph cut-points for the physical activity classification. In a preceding study, we used an indirect calorimeter to analyze the ActiGraph cut-point validity to separate minPA and LIPA in sitting and standing, and noticed a substantial validity gain when using a lower cut-point for sitting (75 cpm, kappa of 0.69) and a higher cut-point for standing (150 cpm, kappa of 0.66) as compared to using the standard 100 cpm for sitting and standing (kappa of 0.56 for sitting and standing) [21]. The same study showed that the 100 cpm cut-point systematically overestimates inactive sitting and underestimates inactive standing [21]. This observation is most likely caused by the fixed waist height while sitting, causing fewer vertical counts in sitting than in standing at the same activity level. A lower cut-point to detect minPA in sitting is also in line with a study by Crouter and colleagues (2013) who noticed a 10% overestimation of the 100 cpm cut-point to detect minPA compared to an indirect calorimeter [30]. However, as a sort of sensitivity analysis to check whether the reported biases were introduced by the posture-specific cut-points, we re-run the entire POPAI classification using a cut-point of 100 cpm for sitting and standing, and noticed only a slightly lower bias for total sedentary time (−2.1% of the waking hours).

POPAI with its separation into inactive and active sitting and standing provides a unique inside into the daily behavior while taking advantage of each of the single sensors’ strengths. POPAI can be expanded to cover the full 24-h spectrum of the day by including bedtime (e.g., with an automated activPal algorithm as in this study used to detect valid waking hours), and it can be further detailed by separating posture and physical activity level in more detail (e.g., separating LIPA and MVPA stepping). An example of such a detailed 24-h analysis is given in Appendix A. However, the most serious limitation of POPAI is the use of two sensors instead of only one as there is currently no single sensor measuring both components of SB simultaneously in an accurate and precise manner. In this regard, we welcome future sensor and algorithm developments to calibrate and validate algorithms against posture and physical activity simultaneously. A method requiring only one sensor to measure both components of SB accurately and precisely would dramatically advance our research area.

### 4.4. Critical Study Appraisal

Since POPAI is based on the two proprietary data processing techniques of the single sensors, it is not independent of the single sensor methods and we have not performed any advanced statistical inference testing. Instead, we used the 95% confidence interval of the Bland-Altman bias to indicate significant effects, but we consider it much more important to look at the actual magnitude of the bias (accuracy) and 95% limit of agreement (precision) to decide whether a single sensor method might have a sufficient accuracy and precision to answer a given research question. The presented analysis excluded bedtime although sleep should have been excluded. However, we are not aware of any valid algorithm to detect sleep with the two sensors used in the study, and thus bedtime was used as a proxy for sleep. Furthermore, the separation into inactive and active standing used an ActiGraph cut-point which separates the activity level at 1.5 metabolic equivalents despite the Sedentary Behavior Consensus Project recommends a threshold of 2.0 metabolic equivalents [31]. However, we are not aware of any evidence justifying a higher threshold for standing than sitting, and strongly believe that the time spent above 1.5 metabolic equivalents should be considered LIPA regardless of body posture. In fact, if we had taken a higher cut-point to separate inactive and active standing, this would have resulted in even more time spent inactive standing and less time spent active standing.

Compared to other samples, our sample accumulated a very similar amount of sitting (activPal data: 58.6% vs. 58.0–58.1% of the waking hours [16,32]), but spent somewhat less time being inactive (ActiGraph data: 55.1% vs. 60.3–62.8% of the waking hours [16,32]). Unfortunately, the posture in which our sample was less inactive remains unknown. In case of sitting, it would mean that our estimate of active sitting is too high for the general population, and thus the true activPal bias would be lower. In case of standing, it would mean that our estimate of inactive standing is too low for the general population, and thus the true ActiGraph bias would be higher. This conclusion is in line with the study by Myers et al. (2017), which included a much more inactive sample (inactive for 70.7% of the waking hours, [28]) and reported a lower relative bias for the posture sensor (11.6%, activPal) but a higher bias for the activity-sensor (29%, SenseWear armband) than this study [28].

The fact that the ActiGraph sedentary estimates and the sedentary accumulation pattern were most often closer to POPAI than the ones of the activPal confirms the results of Fanchamps et al. (2017) but stands in contrast to existing literature recommending the activPal to measure SB [10,11,29]. However, the existing recommendation is primarily based on direct observation, which typically treats sitting as SB regardless of the actual physical activity level, while the present study measured SB with consideration of posture and physical activity. The larger bias of the activPal might partially be explained by the fact that our sample spent more time active sitting (source of activPal bias) than inactive standing (source of ActiGraph bias). Whether this is a generalizable observation remains subject to further studies with a combined posture and physical activity classification.

### 4.5. Practical Implication

This study showed that it seriously matters whether SB is measured with a posture method, a physical activity method, or a combined posture and physical activity method, and none of the single sensor methods can be recommended to measure SB. Following the latest and most widely used definition of SB [15], SB should only be measured with consideration of posture and physical activity. In case such a combined analysis, which currently requires the use of two sensors, is not feasible for whatever reason, the simplest way to circumnavigate the limited accuracy and precision observed in the present study is to talk about sitting (when using a posture sensor) or minPA (when using a physical activity sensor). It should be viewed critical that the definition of the behavior of interest (here: SB) is adapted to the measurement device (here: posture or physical activity sensor), when in fact the measurement device should be adapted to the definition of the behavior of interest. We therefore welcome future sensor and algorithm developments to calibrate and validate single sensor based sedentary measurements against valid reference criteria for posture and physical activity simultaneously. A strict application of the proper terminology will help to collect targeted evidence for SB, for sitting, and for minPA, and makes it possible to figure out which behavior, in which dosage, is responsible for detrimental health effects. A couple of studies already collected data with the two sensors used in this study [13,16,19,32,33], and we hope that the present study might motivate the authors to combine the output of the two sensors to a posture and physical activity based measurement of SB. To simplify such an analysis, we have included the step-by-step instructions for POPAI in Appendix A. If SB is actually harmful to health, we would expect a much stronger relationship between the true amount spent sedentary and detrimental health effects with a combined posture and physical activity classification.

## 5. Conclusions

The present study showed that sitting is not always inactive, and standing is not always active. In fact, the investigated sample spent 24% of the sitting time active, and 41% of the standing time inactive. Accordingly, the activPal measured sedentary time (neglecting active sitting) and the ActiGraph measured sedentary time (neglecting inactive standing) significantly overestimated SB by 30% and 23%, respectively. Future studies should carefully consider this limitation, and ideally combine a posture and a physical-activity sensor to measure SB compliant with its definition when studying associations between SB and health outcomes or interventions to lower SB. The simplification of measuring only one of the two SB components leads to inaccurate and imprecise estimates.

## Figures and Tables

**Figure 1 ijerph-17-08864-f001:**
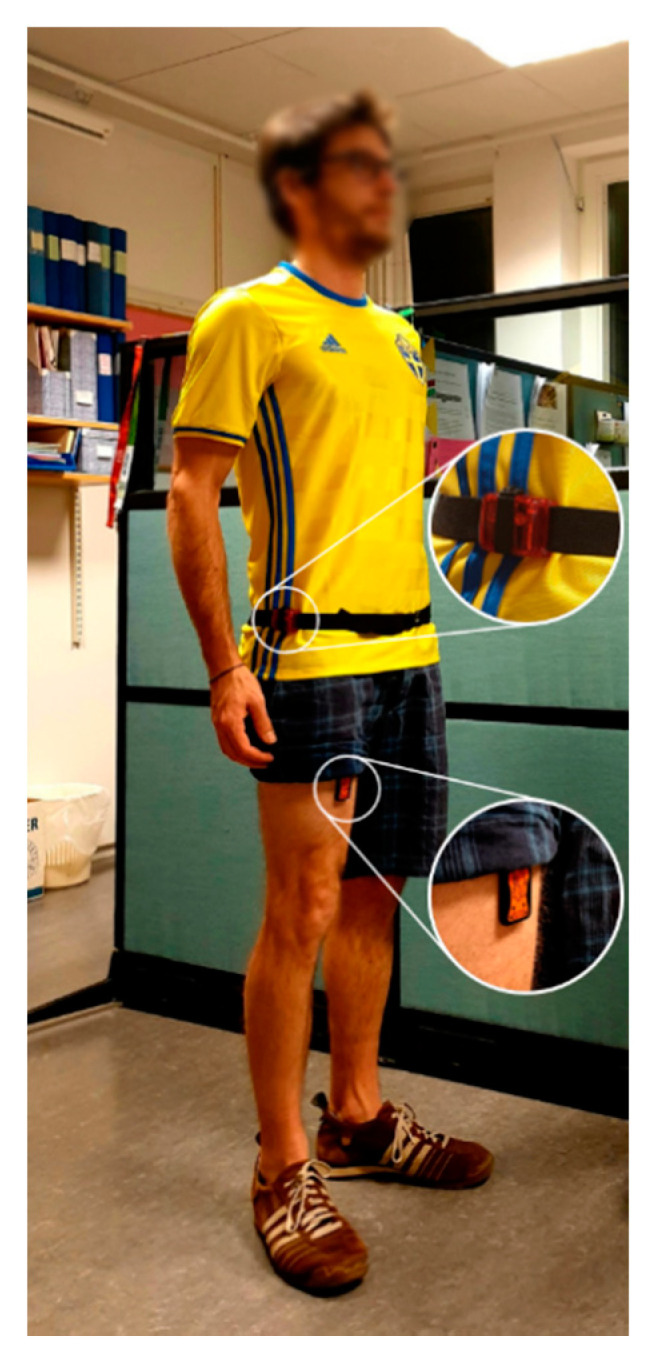
Illustrative figure of a participant wearing the activPal on the right thigh and the ActiGraph GT3X on a belt around the waist. For the measurement, the activPal was packed waterproof and covered with an adhesive patch.

**Figure 2 ijerph-17-08864-f002:**
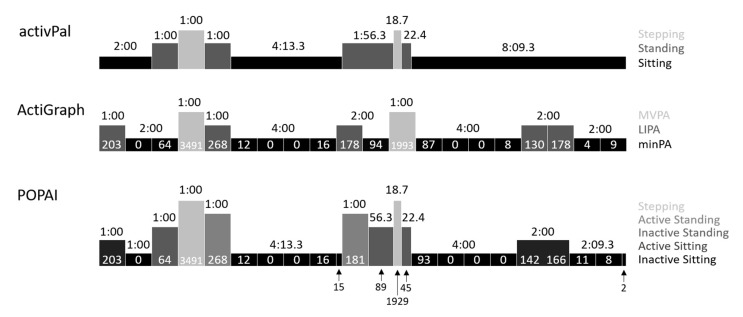
Exemplary classification of a 20-min recording with each method. Indicated is the behavior duration (in minutes:seconds) and the ActiGraph counts. Abbreviations: minimal-intensity physical activity (minPA), light-intensity physical activity (LIPA), moderate- to vigorous-intensity physical activity (MVPA), Posture and Physical Activity Index (POPAI).

**Figure 3 ijerph-17-08864-f003:**
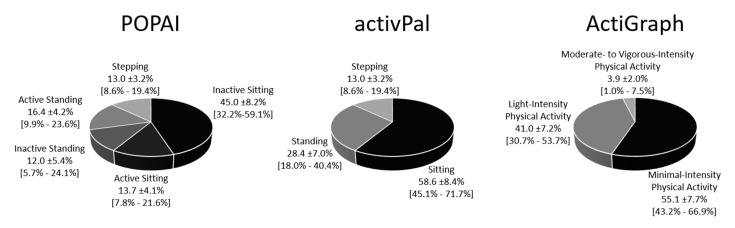
Daily wake time use for each method. Indicated is the mean ±standard deviation [5th to 95th percentile] in percentage of waking hours (100% equals 15.0 h) for each behavior and method, *n* = 100.

**Figure 4 ijerph-17-08864-f004:**
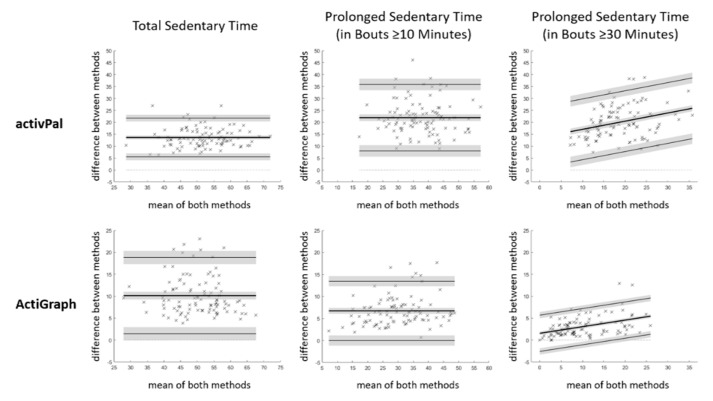
Bland-Altman plots for total and prolonged sedentary time. Data presented in % of waking hours (100% equals 15.0 h, *n* = 100). Indicated is the bias (bold line) and 95% limit of agreement (thin lines), both with 95% confidence interval (in grey). Note that the activPal *y*-axis ranges up to 50% and the ActiGraph *y*-axis up to 25% of waking hours.

**Figure 5 ijerph-17-08864-f005:**
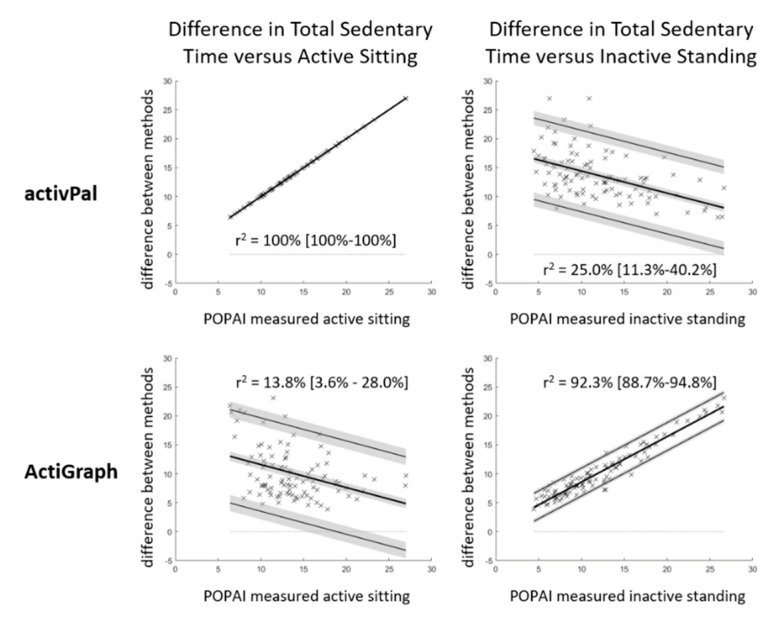
Modified Bland-Altman plots of the bias for total sedentary time versus active sitting and inactive standing. Data presented in % of waking hours (100% equals 15.0 h, *n* = 100). Indicated is the bias (bold line) and 95% limits of agreement (thin lines), both with 95% confidence interval (in grey). The r^2^ indicates the proportion of the bias for total sedentary time that can be explained by active sitting and inactive standing.

**Table 1 ijerph-17-08864-t001:** Bland-Altman comparison of daily sedentary time. Average sedentary time for each method with activPal and ActiGraph bias and limit of agreement (LoA), indicated in % of waking hours. 100% equals 15.0 h, *n* = 100.

	POPAI	activPal			ActiGraph		
Sedentary Time	Mean [95% CI]	Mean [95% CI]	Bias [95% CI]	[95% LoA]	Mean [95% CI]	Bias [95% CI]	[95% LoA]
total	45.0 [43.4–46.6]	58.6 [57.0–60.3]	13.6 * [12.8–14.5]	[5.5–21.8]	55.1 [53.6–56.7]	10.1 * [9.3–11.0]	[1.4–18.8]
in bouts ≥10 min	26.5 [24.8–28.3]	48.5 [46.7–50.2]	21.9 * [20.5–23.3]	[8.0–35.8]	33.3 [31.4–35.2]	6.7 * [6.1–7.4]	[−0.0–13.5]
in bouts ≥30 min	8.6 [7.4–9.8]	28.6 [27.0–30.1]	20.0 *^, #^ [18.7–21.3]	[7.3–32.7]	11.6 [10.3–13.0]	3.1 *^,#^ [2.7–3.5]	[−1.1–7.2]

* significant bias to POPAI based on the 95% confidence interval (CI); # Bland-Altman regression approach was used.

**Table 2 ijerph-17-08864-t002:** Bland-Altman comparison of the sedentary accumulation pattern. Sedentary accumulation pattern for each method with activPal and ActiGraph bias and limit of agreement (LoA), *n* = 100.

	POPAI	activPal			ActiGraph		
	Mean [95% CI]	Mean [95% CI]	Bias [95% CI]	[95% LoA]	Mean [95% CI]	Bias [95% CI]	[95% LoA]
**Number of Sedentary Bouts**							
total [number per day]	66.4 [63.3–69.4]	49.5 [47.3–51.7]	−16.9 *^,#^ [−19.5–−14.2]	[−43.2–9.5]	83.1 [80.5–85.8]	16.8 *^,#^ [15.1–18.5]	[−0.1–33.6]
bouts ≥10 min [number per day]	12.0 [11.4–12.6]	14.9 [14.4–15.5]	2.9 *^,#^ [2.4–3.5]	[−2.4–8.2]	14.6 [14.0–15.3]	2.6 * [2.3–2.9]	[−0.3–5.6]
bouts ≥30 min [number per day]	1.7 ^(np)^ [1.5–1.9]	4.8 ^(np)^ [4.6–5.0]	3.1 * [2.9–3.3]	[0.7–5.5]	2.3 ^(np)^ [2.1–2.6]	0.6 *^,#^ [0.5–0.7]	[−0.3–1.5]
**Further Accumulation Pattern Variables**							
median bout length [minute]	3.0 ^(np)^ [3.0–3.0]	3.8 ^(np)^ [3.5–4.5]	0.9 *^,#^ [0.6–1.1]	[−1.7–3.4]	3.0 ^(np)^ [3.0–3.0]	−0.5 *^,#^ [−0.6–−0.4]	[−1.8–0.8]
% of time spent ≥ median bout length [%]	89.2 [88.8–89.6]	94.0 [93.6–94.3]	4.8 * [4.2–5.3]	[−0.6–10.1]	91.0 [90.5–91.4]	1.7 * [1.2–2.2]	[−3.3–6.8]
half-life bout duration [minute]	12.0 ^(np)^ [11.0–12.8]	28.3 ^(np)^ [26.9–30.0]	17.0 *^,#^ [16.0–18.0]	[7.0–27.0]	12.0 ^(np)^ [11.0–13.0]	0.5 * [0.1–0.8]	[−3.1–4.0]
Gini-Coefficient [unit less]	0.17 [0.16–0.18]	0.22 [0.20–0.23]	0.05 *^,#^ [0.04–0.05]	[−0.05–0.14]	0.17 [0.17–0.18]	0.00 [−0.00–0.01]	[−0.05–0.06]

* significant bias to POPAI based on the 95% confidence interval (CI); # Bland-Altman regression approach was used; (np) non-parametric median with 95% confidence interval is shown (data non-normal distributed).

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
