# Peer review of "Is Sitting Always Inactive and Standing Always Active? A Simultaneous Free-Living activPal and ActiGraph Analysis"

_ijerph, 2020, doi:10.3390/ijerph17238864_

Round 1

Reviewer 1 Report

It is a well done article that only needs to expand the introductory part with bibliography in support of the analyzed constructs.

My comments refer to bibliographic sources because there are few reference articles to support the variables and processes analyzed. The articles are very general the literature must be very supportive of the variables and processes related in the empirical studies. Especially in the introductory part but also in the conclusions part. Furthermore, as regards the results it is appropriate to further detail the results of the areas analyzed.

Publication is strongly recommended.

Reviewer 2 Report

Dear authors,

The article is well structured and well developed. The research question is well explained and can be considered relevant for the journal. The methodology used was well explained, allowing the research to be replicated by other researchers. The content of the results and the discussion needs to be better structured, but it presents relevant material to answer the research question.

I found some aspects to be improved:

- In line 42, you wrote:

“To measure SB reliably and accurately, it is well established that sensor-based methods should be employed, and accelerometers are the method of choice [7,8]. Due to different placement and data processing, these can be separated in two types: sensors to measure posture (also known as inclinometers) and sensors to measure physical activity (also known as accelerometers).

I was quite confused with the phrase “accelerometers are the method of choice” related to the successive phrase “sensors to measure physical activity (also known as accelerometers).” What do accelerometers measure in this context?

- At the beginning of chapter 2 (Materials and Methods), I advise you to introduce a brief description (a paragraph or a sentence) about the methodology used. I do not recommend going directly into the chapter discussing about the participants.

- In line 107, you wrote:

“A detailed discussion of the asynchronous sensor clocks can be found 107 elsewhere [17], an example is given in the figure in supplementary material 1.

I advise you to improve this sentence. It seems unclear to me.

- In line 126, you wrote:

“was classified as SB, light-intensity physical activity (LIPA), or moderate- to vigorous-intensity 126 physical activity (MVPA). Subsequent minutes with the same activity classification were summarized into bouts. Vertical axis cut-points of 100 cpm and 1’952 cpm were used to separate SB, LIPA, and MVPA. Note that SB was renamed to minPA to clarify that only the physical activity component of SB is considered”.

Why not call minPA instead of SB from the beginning?

- Figure 1 and figure 2 have a really long explanation. I suggest introducing it in the body of the text.

- The words in figures 2, 3 and 4 are small. Write them bigger

- Subchapters 3.2 and 3.3 present a very synthetic text, while figures and tables present a lot of data. For a reader it is difficult to understand what data is important to be analyzed. These figures also present a lot of text in their explanation, thus advising to leave it in the body of the text. I advise you to improve these two sub-chapters

- Table 2 is better to leave on the same page

In line 278 and in conclusion, you wrote:

“In fact, 24% of the sitting time was spent active (LIPA), and 41% of the standing time was spent inactive (minPA).”

This is not an absolute measurement. It was extracted from data collected in the researched population. Make that explicit.

- Subchapters 4.1 and 4.2 do not seem to me to be of a discussion section. I do not find any references in them. I would advise to introduce them in the results section, and eventually join them with subchapters 3.2 and 3.3

- I found some references not in number (eg Ellingson and colleagues, or Myers and colleagues)

Best regards

Reviewer 3 Report

This paper present a study to determine the true amount of active sitting and inactive standing in daily life and to analyze by how much the two behaviors falsify the sedentary estimates and the sedentary accumulation pattern of the activPal and ActiGraph compared to a combined sensor analysis, the Posture and Physical Activity Index.  The study included a sample size of 100, with 95% confidence. The result showed that 24% of sitting is active and 41% of standing is inactive, which means that sitting is not always inactive and standing is not always active.  The activPal measured sedentary time and the ActiGraph measured sedentary time significantly overestimated SB by 30& and 23%, respectively. 

Overall, this is a high quality paper.  The overall organization and section structure are consistent with the proposed objectives.  I would consider this work as a good contributor to IJERPH.  Following remarks are suggested to be applied:

  • In Section 2.2, it would be better to have a photo to demonstrate how activPal and ActiGraph GT3X sensors are mounted to human body.
  • In Section 2.3 and 2.4, although authors trying to illustrate the mothed used in time detection and classification, but it is still not clear and may confuse readers. A diagram may help.
